# A randomized controlled trial testing a virtual program for Asian American women breast cancer survivors

Eun-Ok Im [1] ✉, Wonshik Chee [1], Sudeshna Paul [2], Mi-Young Choi[2,3], Seo Yun Kim [2], Janet A. Deatrick [4], Jillian Inouye [5], Grace Ma[6], Salimah Meghani[4], Giang T. Nguyen [7], Marilyn M. Schapira[4], Connie M. Ulrich [4], SeonAe Yeo[8], Ting Bao[9], David Shin[10] & Jun J. Mao [9]

A culturally tailored virtual program could meet the survivorship needs of Asian American women breast cancer survivors (AABC). This study aims to determine the efficacy of a culturally tailored virtual information and coaching/support program (TICAA) in improving AABC's survivorship experience. A randomized clinical trial (NCT02803593) was conducted from January 2017 to June 2020 among 199 AABC. The intervention group utilized TICAA and the American Cancer Society [ACS] website while the control group used only ACS website for 12 weeks. The outcomes were measured using the SCNS-34SF (needs; primary), the MSAS-SF (symptoms; secondary), and the FACT-B (quality of life; secondary). The data were analyzed using an intent-to-treat approach. The intervention group showed significant reductions in their needs from the baseline (T0) to post 4 weeks (T1) and to post 12 weeks (T2). Although the changes were not statistically significant, the intervention group had decreased symptoms from T0 to T2 while the control group had an increase in their symptoms. The intervention group had a significant increase in their quality of life from T0 to T2. A culturally tailored virtual program could therefore improve quality of life in AABC patients. *Trial Registration*: To Enhance Breast Cancer Survivorship of Asian Americans (TICAA), NCT02803593, https://clinicaltrials.gov/ct2/show/NCT02803593?titles= TICAA&draw=2&rank=1

Among Asian Americans, the leading cause of death is cancer, and breast cancer is the most common cancer in Asian American women[1,2]. Also, the incidence rate of breast cancer has increased fastest for Asian Americans among ethnic groups[3]. Yet, Asian American breast cancer survivors delay seeking care, are less likely to report symptoms, and rarely obtain care or support partially because of their cultural values, beliefs (e.g., stigma attached to breast cancer), and language barriers[4–6].Furthermore, they tend to have fewer sources of information and support and worse quality of life[7,8].

[1]The University of Texas at Austin, 1710 Red River St, Austin, TX 78712, USA. [2]Emory University, 1520 Clifton Road, Atlanta, GA 30322, USA. [3]Chungbuk National University, 1 Chungdae-ro, Seowon-gu, Cheongju-si, Chungcheongbuk-do, South Korea. [4]University of Pennsylvania, 418 Curie Blvd, Philadelphia, PA 19104, USA. [5]University of Hawaii, 2528 McCarthy Mall, Webster Hall 402, Honolulu, HI 96822, USA. [6]Temple University, 1801 N Broad St, Philadelphia, PA 19122, USA. [7]Harvard University Health Services, 75 Mt. Auburn Street, Cambridge, MA 02138, USA. [8]University of North Carolina, Carrington Hall, S Columbia St, Chapel Hill, NC 27599, USA. [9]Memorial Sloan Kettering Cancer Center, 321 East 61st Street, Room 456, New York, NY 10065, USA. [10]University of California, Los Angeles, 855 Tiverton Dr, Los Angeles, CA 90024, USA. ✉e-mail: eunok.im@austin.utexas.edu

The COVID-19 pandemic has recently accelerated telemedicine and virtual-delivered interventions in oncology care[9,10]. A technology-based virtual program through computers and mobile devices could potentially meet the need of vulnerable populations with increased accessibility while reducing the cost of the intervention in expensive and busy health care settings[11–13]. Moreover, a technology-based virtual intervention without face-to-face in-person interactions could provide anonymity for the women whose cultures stigmatize breast cancer. Despite the high potential, very few virtual interventions have been researched for ethnic minority breast cancer survivors, especially for Asian Americans[14,15]. Using cultural tailoring based on individual sub-ethnic groups' cultural attitudes, we developed and pilot tested a virtual information and coaching/support program without in-person face-to-face interactions among this specific population[16].

This study aimed to determine the efficacy of a technology-based (virtual) information and coaching/support program in reducing unmet needs, reducing symptom distress, and improving quality of life among Asian American women breast cancer survivors (TICAA).

## Results

### Participants' flow

From January 2017 to June 2020, a total of 843 Asian American breast cancer survivors consented to participate. Among those who were screened to be eligible, only 268 women completed the baseline questionnaire; others were not contactable due to multiple reasons (e.g., moving out of country, busy with treatments). This attrition rate is similar to that of longitudinal online trial studies[17]. Then, an additional 69 women were excluded because they were not eligible because of several reasons (e.g., missing information on primary outcomes at the baseline). Thus, a total of 199 Asian American breast cancer survivors (104 in the intervention group and 95 in the control group) were included in the final data analysis (Fig. 1).

### Participants' characteristics

Only women were included (see the Supplementary Data 1). The average age was 52.35 years (standard deviation [SD] = 12.51); and 53.0%, 22.9%, and 24.1% of the participants were Chinese Americans, Korean Americans, and Japanese Americans, respectively. About 40.1% were employed, and about 62.7% perceived their family income as sufficient. About 91.0% had access to healthcare, and about 87.8% were not born in the U.S. The average years since the diagnosis of breast cancer was 3.65 years (SD = 3.88). About 43.0% had Stage 1 breast cancer, and about 37.8% had Stage 2 breast cancer. At T0, the average score of the needs was 83.81 (SD = 29.47), the average symptom distress score was 0.85 (SD = 0.57), and the average quality of life score was 99.01 (SD = 22.33). There existed no significant differences between the two groups except access to health care ($X^2 = 5.17$, $p < 0.02$) and age at immigration ($t = -2.34$, $p < 0.02$); these two variables were controlled in the subsequent data analysis process.

### Primary outcome—Needs

Within the generalized estimating equations model, the SCNS scores (needs) showed a significant time effect only. Over time, the intervention group had significant decreases in the SCNS scores from the baseline (T0) to post 4 weeks (T1) and from T0 to post 12 weeks (T2) (see Table 1 and Fig. 2). Although the control group had decreases in the SCNS scores from T0 to T1 and to T2, the decreases were not statistically significant (see Table 1 and Fig. 2).

### Secondary outcomes—Symptoms and quality of life

Within the generalized estimating equations model, no significant time, group, or time × group interaction effects were found for the MSAS-SF scores (symptoms). Although the changes were not statistically significant, the intervention group had decreases in the MSAS-SF

scores from T0 to T2 while the control group had increases in the MSAS-SF scores (see Fig. 3).

Within the generalized estimating equations model, marginally significant time × group interaction effects were found in the FACT-B scores (quality of life) from T0 to T2 ($p = 0.06$; see Table 1). Compared to the control group, the FACT-B scores of the intervention group significantly increased from T0 to T2 ($\beta = 5.25$, 95% CI, 0.43–10.08, $p < 0.05$; see Fig. 4).

## Discussions

The findings indicated that, over time, those who used TICAA and ACS websites showed significant within group reduction in their needs from the baseline to post 4 weeks and to post 12 weeks. Compared with those who used the ACS website alone, those who used TICAA and ACS website had significant increases in their quality of life from the baseline to post 12 weeks, but no significant changes in symptoms. This study supports that a 12-week culturally tailored virtual intervention using computers and mobile devices could improve the survivorship experience of Asian American breast cancer survivors. The findings of this study are consistent with the literature reporting that virtual coaching and support programs could change health behaviors, and consequently improve health outcomes[18,19]. Prior research suggests that technology-based interventions potentially could overcome geographical barriers and reduce time and travel burden for participants, with a particular advantage of reaching out to marginalized groups such as racial/ethnic minorities, especially with stigmatized conditions[20]. Yet, this study found that both TICAA with the ACS website and the ACS website alone decreased the needs of the participants within 4 weeks although the decreases were not statistically significant. The finding that the ACS website alone decreased the participants' needs supports the effectiveness of the attention control condition used in this study—the ACS website. This finding could mean that a simple use of an informational website could meet the needs of Asian American breast cancer survivors.

This study also found that TICAA with the ACS website had larger decreases in the needs of the participants from the baseline to 4 weeks and slight increases from 4 weeks to 12 weeks. Also, the study found that those who used TICAA and ACS had larger increases in the quality of life from the baseline to 4 weeks and smaller decreases from 4 weeks to 12 weeks. These findings could be interpreted in several ways. First, it could mean that a short intervention period (4 weeks in this study) would work better for this type of virtual interventions to decrease Asian American breast cancer survivors' needs and improve their quality of life. In the literature, 12 weeks are a typical period of technology-based interventions[21]. However, these findings on larger changes in the needs and quality of life at 4 weeks might indicate that 4 weeks would be a better intervention period for this type of virtual interventions for Asian American women breast cancer survivors. Yet, the finding on changes in symptoms was a little bit different; although the changes were not statistically significant, the decreases in symptom distress were prominent at 12 weeks compared with at 4 weeks. We hypothesize that once survivors learned information about what they could do, it would require time for them to seek out care and then allow the strategies (e.g., medications, psychotherapy, or acupuncture) to reduce their specific symptoms. Alternatively, they may have experienced new symptoms over time and are employing the resources from the intervention.

The finding that the effects on the symptom scores were not statistically significant over time and between groups could be interpreted in several ways. First, it could mean that symptom distress might not be easily changed during a short intervention period as discussed above, which would require a different approach than a short-time technology-based information and coaching/support program. Second, MSAS-SF might not be sensitive enough to measure the symptom distress among breast cancer survivors because many of the

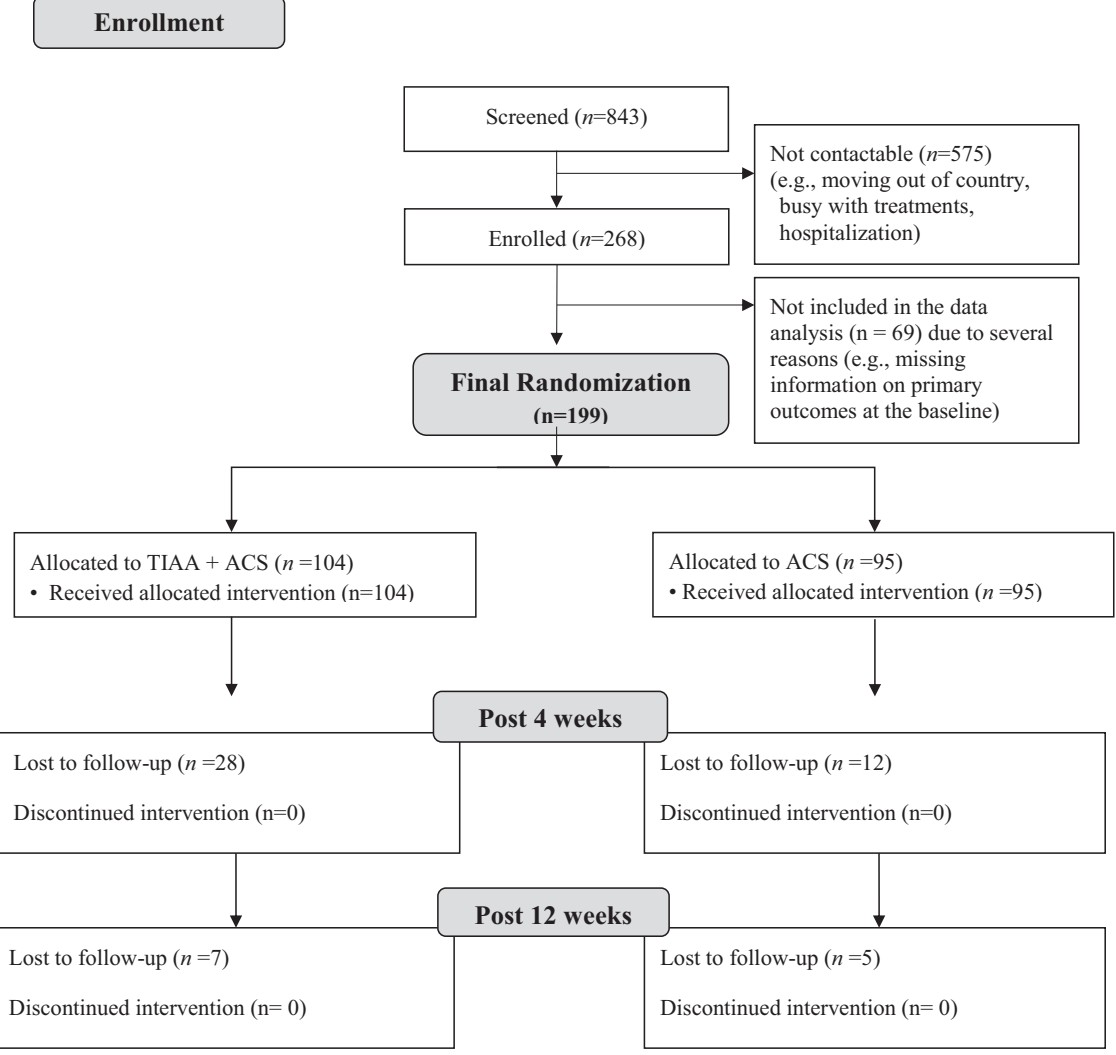

**Fig. 1 | Participant Flow Diagram.**

## Table 1 | Changes in the outcome variables by group and time (*N* = 199)

| Outcomes | Time | Control | | Intervention | | *P*ᵃ values | | |
|---|---|---|---|---|---|---|---|---|
| | | Mean | SE | Mean | SE | Group | Time | Group*time |
| *Primary outcome* | | | | | | | | |
| SCNS scores (Needs) | Pre | 79.02 | 3.05 | 82.50 | 3.23 | 0.50 | | 0.72 |
| | Post1 | 73.83 | 2.95 | 74.81 | 3.22 | | 7.61E−05 | |
| | Post2 | 72.21 | 3.10 | 75.85 | 3.17 | | | |
| *Secondary outcomes* | | | | | | | | |
| MSAS scores (Symptom Distress) | Pre | 0.67 | 0.06 | 0.91 | 0.06 | 0.04 | 0.58 | 0.32 |
| | Post1 | 0.77 | 0.06 | 0.90 | 0.07 | | | |
| | Post2 | 0.76 | 0.07 | 0.84 | 0.08 | | | |
| FACT-B scores (Quality of Life) | Pre | 104.92 | 2.09 | 97.27 | 2.59 | 0.17 | 0.17 | 0.06 |
| | Post1 | 104.11 | 2.18 | 101.87 | 2.55 | | | |
| | Post2 | 104.90 | 2.08 | 101.94 | 2.73 | | | |

Model: (Intercept), Group, Time, Access to healthcare, Age at immigration, and Group*Time.
*SE* Standard Error, *SCNS* Support Care Needs Survey, *MSAS* Memorial Symptom Assessment Scale, *FACT-B* Functional Assessment of Cancer Therapy Scale-Breast Cancer.
ᵃGeneralized estimating equations adjusted for access to healthcare and age at immigration.

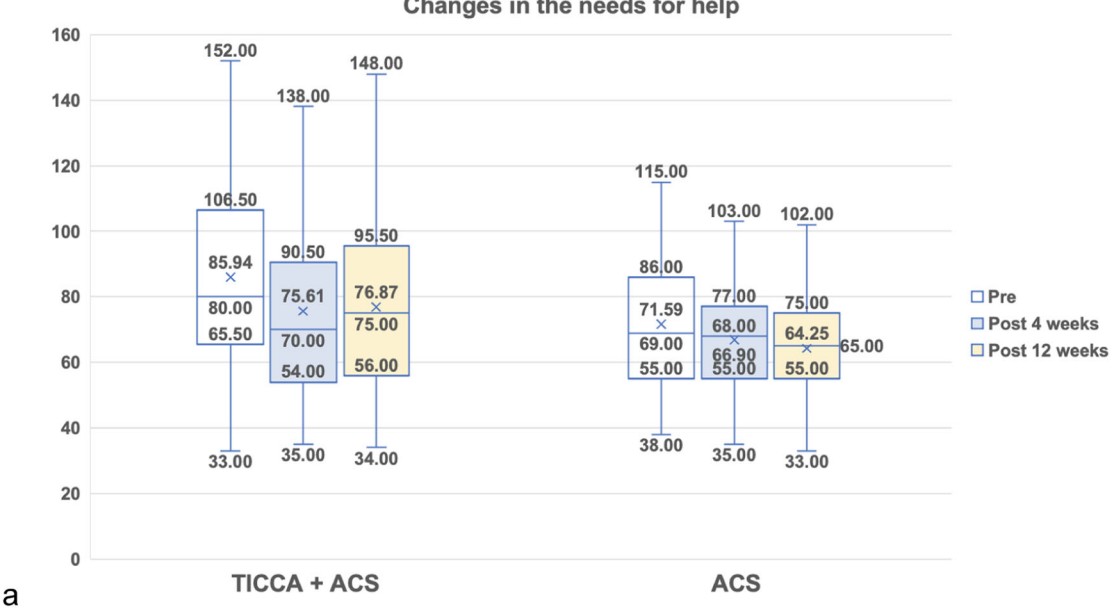

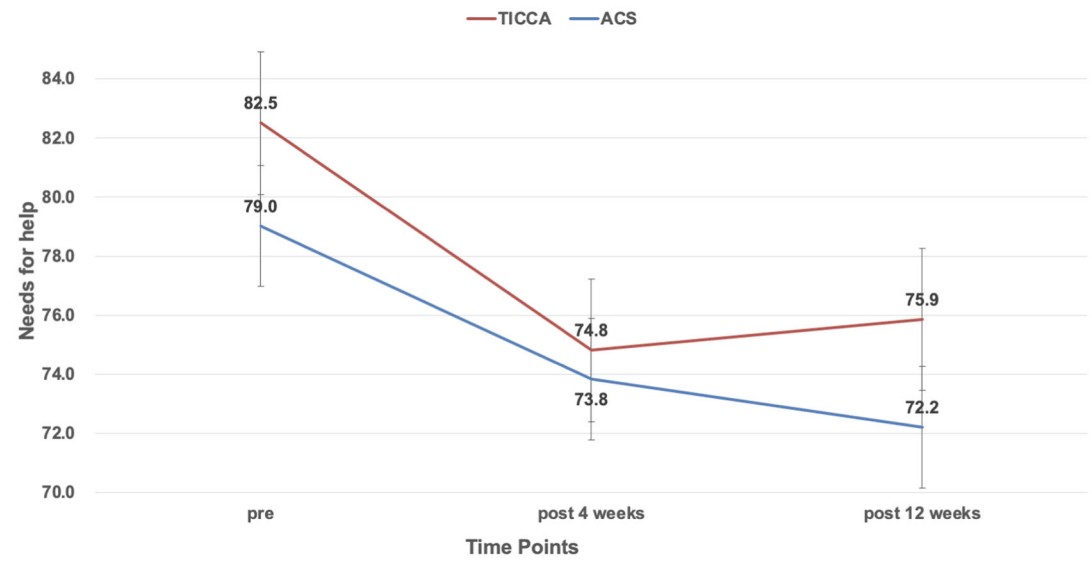

**Fig. 2 | Changes in the needs for help over 12 weeks.** The number of samples in the TICCA + ACS group was 103 at baseline, 76 after 4 weeks, and 71 after 12 weeks. The number of samples in the ACS group was 88 at baseline, 78 after 4 weeks, and 66 after 12 weeks. **a** Box plots defined a graphical exploratory analysis of the changes in the needs for help by group. The median, quartiles, maximum, and minimum values are presented in each box plot at each time point. **b** The mean change in the need for help over time is presented along with standard error bars.

items measure acute symptoms such as nausea/vomiting that would be applicable only to those undergoing treatments. Finally, the mean symptom distress scores of both groups at the pre-test were too low and created a floor effect that made it difficult to detect any intervention effects.

This study has several limitations. First, there was a sizeable segment of the recruited participants who were not contactable after successfully going through the screening process; thus, there could exist some potential selection biases as typically seen in clinical trials[22]. This study required the participants to have regular access to the Internet, which might limit its generalizability to those with poor digital literacy or access; the participants of this study tended to be highly educated. Furthermore, considering that the data collection period included

6 months of the COVID pandemic, there could exist some influences of other contextual factors that were not measured. Also, the ACS website has rich information about resources and support groups; therefore, our control is active, which could limit our ability to detect important effects beyond typically seen in routine care. Indeed, those who used ACS alone had decreases in the needs by 12 weeks (although the decreases were not statistically significant). Fourth, the study did not include a long-term follow up. Fifth, the study did not have the information on diagnosis or treatment trajectories of individual participants that could influence the intervention efficacy. Rather, the study included the type and stage of breast cancer, time duration after the diagnosis of breast cancer, and treatment modalities that the participants went through, which were considered in the data analysis process. Lastly,

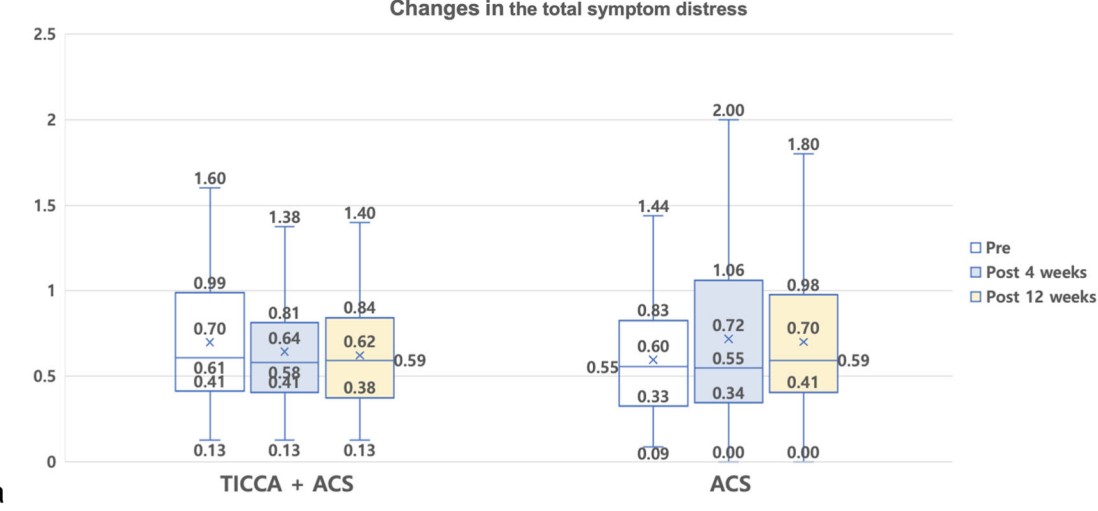

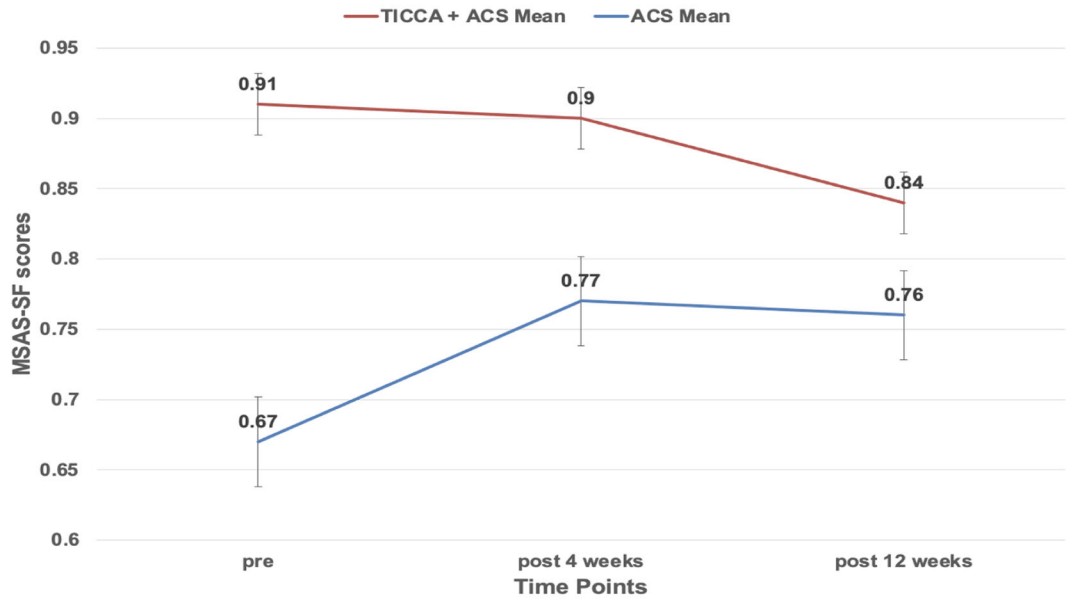

**Fig. 3 | Changes in the total symptom distress over 12 weeks.** The number of samples in the TICCA + ACS group was 103 at baseline, 76 after 4 weeks, and 71 after 12 weeks. The number of samples in the ACS group was 88 at baseline, 71 after 4 weeks, and 66 after 12 weeks. **a** Box plots defined a graphical exploratory analysis of the changes in the total symptom distress by group. The median, quartiles, maximum, and minimum values are presented in each box plot at each time point. **b** The mean change in the total symptom distress over time is presented along with standard error bars.

although the Asian American community is ethnically diverse, we enrolled the participants from only three sub-ethnicities, which could limit its generalizability to other subgroups. In addition, this study was limited to Asian Americans who identified as women and therefore did not include Asian American breast cancer survivors of other genders and gender identities. Despite the limitations, this study supported that a technology-based (virtual) information and coaching/support program was effective in improving Asian American women breast cancer survivors' quality of life. The study provides several implications for future technology-based (virtual) research and practice among Asian American women breast cancer survivors.

## Methods

This study was approved by the Institutional Review Board of Emory University. Electronic informed consent from all participants were obtained.

### Settings and samples

The recruitment settings included both online and offline cancer support groups and communities/groups for Asian Americans across the U.S. A total of 1313 cancer support groups and communities/groups were contacted, and 314 among them actually posted the study announcements. Three sub-ethnic groups of Asian Americans were selected: Chinese Americans, Korean Americans, and Japanese Americans. Chinese Americans are the biggest sub-ethnic group[23], Korean Americans are the fastest growing sub-ethnic group[23], and Japanese Americans are the sub-ethnic group with the highest incidence rate of breast cancer[24].

Participants were self-reported Asian American women aged 21 years and older who identified their sub-ethnicity as Chinese Americans, Korean Americans, or Japanese Americans; had a breast cancer diagnosis; could read and write English, Mandarin Chinese, Korean or Japanese; and had access to the Internet through computers or mobile

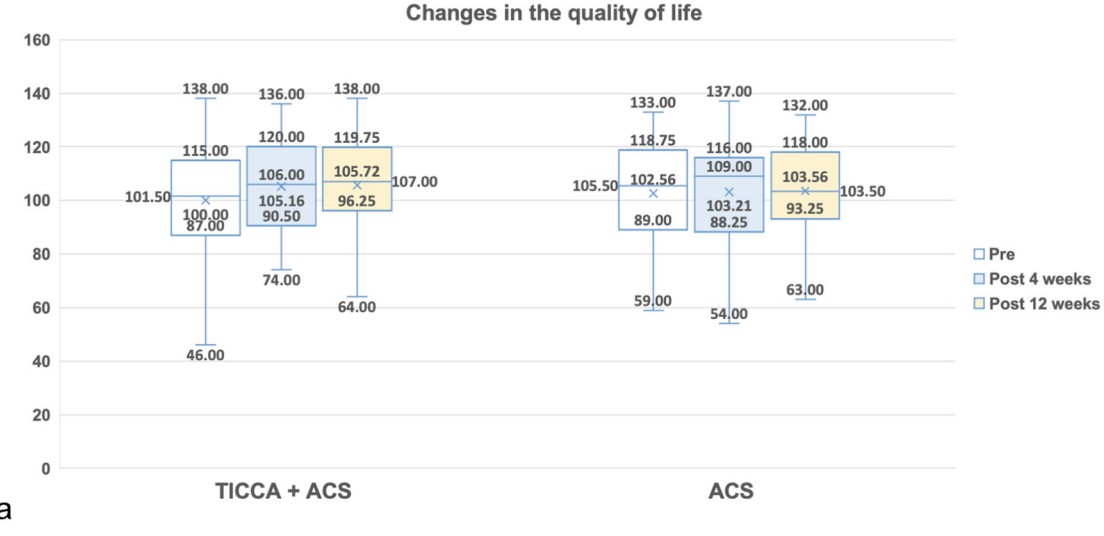

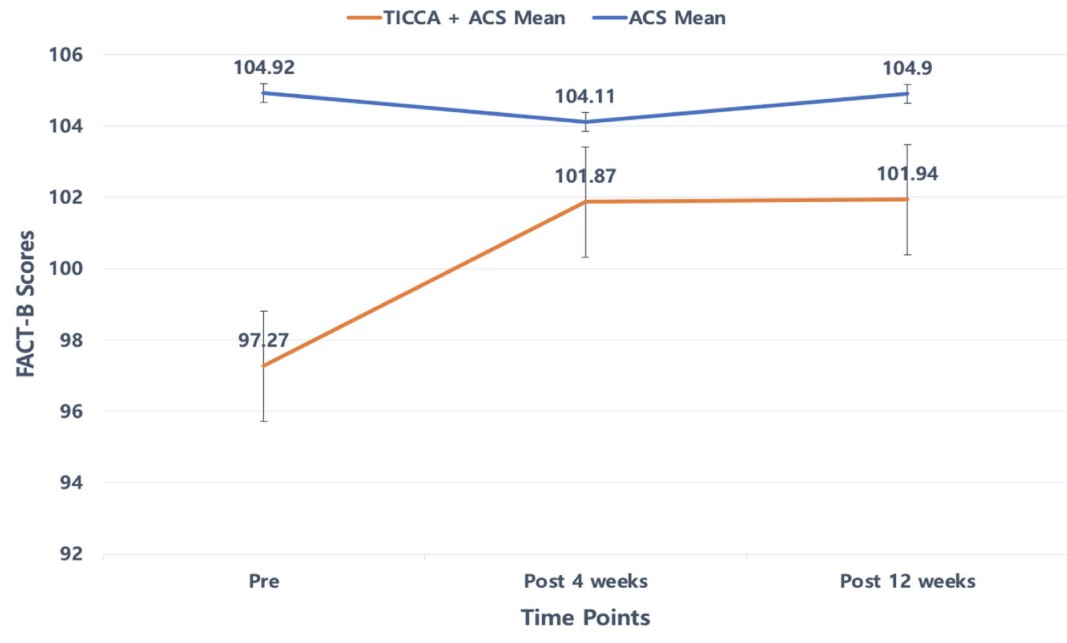

**Fig. 4 | Changes in the quality of life over 12 weeks.** The number of samples in the TICCA + ACS group was 103 at baseline, 76 after 4 weeks, and 71 after 12 weeks. The number of samples in the ACS group was 88 at baseline, 81 after 4 weeks, and 69 after 12 weeks. **a** Box plots defined a graphical exploratory analysis of the changes in the quality of life according to group. The median, quartiles, maximum, and minimum values are presented in each box plot at each time point. **b** The mean change in the quality of life over time is presented along with standard error bars.

devices (smart phones and tablets). Originally, only those who had a breast cancer diagnosis within 5 years were included, but the criterion was later changed with the NCI approval to eliminate the time limit due to difficulties in recruitment. When a volunteer met the inclusion criteria, she was stratified by sub-ethnic group, automatically assigned a serial number, and randomized into two groups (an intervention group and a control group) in each sub-ethnic group using an automated random number generator accessible through the website.

**Enrollment and randomization**

When potential participants visited the project website after seeing the study announcement through online and offline cancer support groups and communities/groups for Asian Americans (the study announcement included the website address), they were asked to review the "informed consent." When participants clicked "I agree to participate," they had given their consent. After checking them against the inclusion criteria and quota requirements, only those who met the criteria and requirements were automatically given a serial number separately in each sub-ethnic group and randomized into two groups in each sub-ethnic group using an automated random number generator accessible through the website. The women were then asked to fill out the questionnaire (T0) and provided with IDs and passwords that were randomly assigned by the researchers. Both groups were provided with the link to the ACS website and were asked to use the website for 12 weeks. Both groups were also asked to maintain their usual information searches through existing resources. The intervention group used TICAA for 12 weeks. The research team sent biweekly reminders and thank-you emails to both groups. The reminders

encouraged both groups to use TICAA and/or the ACS website; reminders are an elementary approach to boost adherence in behavioral intervention studies[25]. Women were asked to complete the same questionnaires (excluding the questions on background characteristics) at T1 and T2.

## Control: ACS website
The participants in the control group were directed to use the ACS website on breast cancer through the project website. The ACS website included information on treatment, prevention, genetics, causes, screening, testing, coping, clinical trials, research findings, and statistics related to breast cancer in multiple languages (including Asian languages).

## Intervention: TICAA and ACS
TICAA was guided by the Bandura's Self-Efficacy Theory of Behavioral Change[26]. TICAA was a 12-week intervention. According to the theory, addressing individuals' attitudes, self-efficacy, perceived barriers, and social influences changes their health behaviors and subsequently results in better health outcomes. TICAA consisted of three intervention components that were provided in five languages (English, Mandarin Chinese [Simplified and Traditional], Korean, and Japanese): (a) social media sites; (b) interactive online educational sessions; and (c) online resources. The social media sites provided a medium by which participants could connect to each other and share their own breast cancer survivor experience with peers, have interactions with and get support from peers, and obtain coaching/support from interventionists and peers who were culturally matched[27]. In each site, cultural tailoring was done through incorporating both general and sub-ethnic-specific materials to discuss and culturally appropriate examples from the literature (e.g., cultural taboo) and the qualitative findings of previous studies by the research team[28-33]. Each site consisted of a social media function (like the timeline of *Facebook*), a chat function with a culturally matched bilingual interventionist, and a symptom log function. Fifteen interactive online educational sessions provided information on breast cancer survivorship (general and sub-ethnic-specific topics). These sessions provided correct and updated information on breast cancer and treatment/management strategies so that stigmatization could be reduced by correcting misinformation. Culturally relevant content (e.g., Red Ginseng, herbal medicine, Acupuncture, etc.) was also incorporated into the sessions. Finally, sub-ethnic-specific online resources included 35 online links that were connected to information and/or resources on breast cancer survivorship in English, Mandarin Chinese (Simplified & Traditional), Korean, and/or Japanese, and they were from scientific authorities and health organizations/institutes (general and sub-ethnic-specific).

## Outcomes and co-variates
**Primary outcomes: needs.** The Support Care Needs Survey-34 Short Form (SCNS-34SF)[34] was used to measure the needs of the participants. The SCNS-34SF included 34 items in five domains (10 on psychological; 11 on health systems and information; 5 on physical and daily living; 5 on patient care and support; and 3 on sexuality needs). Each item was on a 5-point scale (1 = *no need for help*, 5 = *high need for help*). All items were summed to obtain the SCNS score (34–170). The scale's Cronbach's α was 0.97 in this study.

**Secondary outcomes: symptom distress.** The Memorial Symptom Assessment Scale-Short Form (MSAS-SF)[35] was used to measure symptom distress (psychological [4 items] and physical symptoms [28 items]) .The MSAS-SF included 32 items on symptoms experienced during the past 7 days (on a 5-point Likert scale; 0 = no symptom to 4 = very much). The MSAS-SF included the Global Distress Index (4 psychologic and 6 physical symptoms), the physical symptom distress score (12 items), the psychologic symptom distress score (6 items), the

total symptom distress, and the number of total symptoms. In this study, the MSAS-SF score was calculated by averaging the distress scores of 32 items. In this study, the scale's Cronbach's α was 0.91

**Secondary outcomes: quality of life.** The Functional Assessment of Cancer Therapy Scale-Breast Cancer (FACT-B)[36] was used to measure the quality of life. The FACT-B included 37-items in 5 domains: physical well-being (7 items), social/family well-being (7 items), emotional well-being (6 items), functional well-being (7 items), and a breast-cancer subscale (BCS) (10 items). The items were on a 5-point Likert scale (0 = not at all to 4 = very much), indicating the interference in patient's lives. In this study, individual item scores were summed to generate the FACT-B score (range, 0–148). In this study, the scale's Cronbach's α was 0.93.

**Sociodemographic and clinical variables.** We measured 14 sociodemographic variables including age, education, religion, marital status, employment, family income, degree of difficulty paying for basics, access to health care, geographical region, urban/rural residence, sub-ethnicity, country of birth, length of stay in the U.S. (years), age at immigration (years), and 5 questions on the level of acculturation (5-point Likert scale; 1 = exclusively own ethnic group to 5 = exclusively American). The questions on the level of acculturation were adopted and modified from the Asian Self-Identity Acculturation Scale[37] to measure the degree of acculturation in multiple ethnic groups. We also measured clinical variables including type and stage of breast cancer, time since diagnosis, and treatment modalities (e.g., radiation therapy, chemotherapy, surgery, hormone therapy). All these questions were used in previous studies of the research team[16,38-42].

## Statistics & reproducibility
This is a randomized controlled trial. The eligible participants were randomized into two groups: (a) those using 12 weeks of TICAA with ACS website on breast cancer and (b) those using 12 weeks of the ACS website alone. The Investigators were not blinded to allocation during experiments and outcome assessment. The data were collected at three time points (T0, T1, & T2).

In the sample size calculation, a sample size of 99 participants in each group (total $N = 198$), with 3 repeated measurements obtained from each participant, would be adequately powered (80%) to detect a slope difference of 0.27, based on a two-group two-level hierarchical design. The calculation assumed a standard deviation of 1, a correlation of 0.1 between observations on the same subject, and an alpha level of 5%. The effect size was conventionally determined based on pilot studies.

An intent-to-treat approach was used; the participants were analyzed in their original randomized conditions regardless of the usages of TICAA or follow-ups missed. Missing values were not substituted. Again, 69 women were excluded during the data analysis process because they were not eligible because of several reasons (e.g., missing information on primary outcomes at the baseline). Because the outcome variables showed non-normal distributions, generalized estimating equations were used to examine group differences in continuous outcomes at each time point, after adjusting for the baseline outcome and covariates that demonstrated differences. There existed no significant differences between the two groups except access to health care ($X^2 = 5.17$, $p < 0.02$) and age at immigration ($t = -2.34$, $p < 0.02$); thus, these two variables were controlled in the subsequent data analysis process. GEE was applied to confirm the differences in the changes of major variables between the two groups over time. In the data analysis process, two variables (access to healthcare and age at immigration) that showed differences in the homogeneity tests of the two groups were entered as covariates and adjusted for the subsequent analyses. Treatment effects were quantified as the mean differences between T1 and T2. The changes in the

average scores of individual outcome variables from T0 to T1 and T2 were calculated in both intervention and control groups.

## Reporting summary

Further information on research design is available in the Nature Portfolio Reporting Summary linked to this article.

## Data availability

The raw data are protected and are not available due to data privacy laws. Also, the data cannot be shared in a public repository because the research team did not get the permission from NIH or from the participants during the research process. The processed deidentified data are available according to the below process. However, any researchers could request the data sharing through contacting PI (Dr. E-O.I) and the data will be shared according to the following procedures that was approved by NIH at the time of the grant award. First, the deidentified data and the associated codebook that defines the data will be available for sharing with other researchers. The data will be available for secondary analyses especially by those who wish to investigate the effectiveness of a culturally tailored technology-based information and coaching/support program in various variables other than our major outcome variables. Any researcher who wishes to use the data must request permission to conduct secondary analyses of the data from PI of the study (Dr. E-O.I.) by e-mail or regular mail and provide PI with a 1-page long abstract (single-spaced) of the proposed analysis and his/her CV. The decision on data sharing will be made by the research team, including the PI, Co-Investigators, and consultants, after they review the abstract and CV. When the research team decides to share the data with the researcher, the data in SPSS format, abstract, and original findings will be provided to the researcher. The researcher will be requested to: (a) agree that she/he will provide the findings from her/his analyses to the PI at the completion of the analyses, (b) acknowledge the original study and the NIH in her/his future publications, and (c) not use the findings from the data for any commercial purposes. This agreement will be made in a written form. The data will have no identifying information to link a subject to her data. The data will be shared to the researcher through OneDrive. The data will be available for 10 years after the completion of the study. Source data are provided with this paper.

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

## Acknowledgements

The study was funded by the National Cancer Institute (NIH/NCI; R01CA203719). Drs. Mao and Bao are supported in part by a grant from the National Institutes of Health/National Cancer Institute Cancer Center (P30 CA008748). We appreciate the efforts made by dozens of research assistants and community consultants during the research process. More importantly, we appreciate the contributions made by our research participants.

## Author contributions

All authors agreed on the final version and met at least one of following criteria (substantial contributions to conception and design; acquisition of data, or analysis and interpretation of data; drafting the paper or revising it critically for important intellectual content).

## Competing interests

The authors declare no competing interests.
