## [Peer Review File · Nature Communications]

REVIEWER COMMENTS

Reviewer #1 (Remarks to the Author): expertise in biostatistics and epidemiology

The authors present their results from a randomized trial to assess the effectiveness of a culturally tailored virtual information, coaching, and support program for Asian American survivors of breast cancer. Within the intervention group, the authors demonstrated a reduction in unmet support care needs over 1- and 3-month periods. Across groups, they demonstrated an increase in quality of life but no difference in symptom distress. Strengths of the work included a well-written manuscript, carefully-developed intervention, intent-to-treat analysis, and most importantly, an effort to address a significant need for reducing important disparities in care-seeking, symptom reporting, and quality of life among Asian American breast cancer survivors. However, substantial weaknesses detracted from the impact of the work, primarily related to the methodology. Further, generalizability of the findings may be limited given that the average time since diagnosis for participants was over 3.5 years.

Major comments:

- It is difficult to define or understand what "use" of the TICAA and ACS websites means in practice across participants, as even though they were given access to the resources, actual usage was not tracked. Was there any assessment if the ACS website was part of the usual/existing resources for participants? Were biweekly reminders and thank you emails sent to both groups or just the intervention arm? A lack of details makes it difficult to assess these potentially important components for interpretation of findings.
- Details of the recruitment settings are lacking, making assessment of potential selection biases (e.g. self-selection) and external validity difficult. The supplement outlines the original plan/protocol but does not detail the various support groups recruited from in the end, any deviations from the original plan, etc.
- The relative timing of where participants were in their treatment/survivorship journey is a critical piece of data compared across groups. Because most survivors were on average 3.5+ years out, the authors should comment on generalizability to women more recently diagnosed.
- Further elaboration on the cultural tailoring methodology would be helpful (e.g. were these informed only by the literature as stated or also through focus groups, community advisory panels, etc.).
- A vague comment is made about adjusting for "covariates that demonstrated differences." More detail about covariate selection and analytic adjustment is needed.
- There is a missed opportunity to statistically compare the full T0-T1-T2 trajectories for the various outcomes across intervention and control groups.
- The rationale for selecting an alpha of 0.10 is weak.

Minor comments:

- It is unclear from the "settings and samples" section if randomization was conducted within sub-ethnic groups (i.e. post-stratification) to achieve similar subethnic group sizes across arms. Also, it is unclear in the main text how participants were directed to the project website.
- The data availability and sharing plan is informal with unclear criteria for approval of requests.
- First sentence of the abstract is not a complete sentence.
- Typo line 102: should read "The COVID-19 pandemic" or simply "COVID-19"
- Reference needed for the second sentence of "limitations" in 305-308.

Reviewer #2 (Remarks to the Author): expertise in clinical trials in breast cancer

The Authors reported the results of a randomized study supporting the use of a technology-based (virtual) information and coaching program in improving breast cancer survivors' quality of life. The study population was composed of Asian American women. Authors acknowledged strengths and limitations of the study but, overall, the discussion is well balanced.

Reviewer #3 (Remarks to the Author): expertise in the cancer experience in under-represented

populations

I find it an important topic to study because we live in a diverse society and our information is not enough tailored to patients' needs.

However, I do have some important comments about the study, which I will sum up here:

- The efficacy of culturally tailored programs has been studied before (e.g. recently doi: 10.1007/s13187-021-01970-y & 10.1177/1054773818755836). I miss here the specific reason why this needed to be studied, why this study is novel among all other studies that evaluated culturally tailored programs.
- Line 95: references 2-4: I would highly suggest to use more recent numbers (see doi: 10.3322/caac.21754)
- Line 101: reference are not peer reviewed articles and the second webpage could not be found. I do believe, however, that COVID has increased racism towards the Asian population, but I do not think this validates the need for this study, which was mostly done BEFORE the pandemic. The authors also mention COVID in their discussion and limitation, but the impact of COVID cannot be enourmous as the trial was finished in June 2020. And maybe the emphasis (during March and June 2020) would lie more on the anxiety of the possible impact of a COVID infection during cancer (treatment). It's therefore maybe better not to include data that was retrieved during the pandemic.
- Lines 113-114: "reducing the unmet needs..." Is this the case, are there unmed needs, symptom distress and a lower quality of care; and if so, it is higher in comparison to other breast cancer patients? It is not because they seek less care, or report less symptoms, they are experiencing unmet needs. The authors need to elaborate on this matter a bit more to my opinion.
- Lines 132-141: repetition of the same information. The also do this between lines 281-291.
- Line 146: why did they send reminders. This probably also has influenced the behavior of participants.
- Sociodemographics: unequal group sizes and the no differences were made in the group > high school education. I would expect to - especially for a study that looks at unmet needs etc, that they would include more groups for education: no education and primary school. Further, they did not ask how and where participants looked for information before the study.
- I would suggest to ask for the calculation of their sample size! I have some doubts about this.
- Lines 237-238: there were some participants lost in follow up, see fig 1 but this wasn't mentioned in the paper. So the total sample size was even lower.
- Line 262: what is FACT-G?
- Line 268: references are outdated.
- Discussion: cancer is a longterm process with often many bad news conversations (diagnosis, treatment that does not work, ...) and different treatment approaches. So what does an intervention of 12 weeks mean? What is the added value of such a program compared to live interventions?

To my opinion there is still some work to be done before this paper can be considered for publication.

If the editors feel that the abovementioned comments can be shared with the authors, they are free to do so.

RESPONSE TO REVIEWERS' COMMENTS

Reviewer #1	Response
Strengths of the work included a well-written manuscript, carefully-developed intervention, intent-to-treat analysis, and most importantly, an effort to address a significant need for reducing important disparities in care-seeking, symptom reporting, and quality of life among Asian American breast cancer survivors.	Thanks.
Generalizability of the findings may be limited given that the average time since diagnosis for participants was over 3.5 years.	This has been added to the limitation section.
It is difficult to define or understand what “use” of the TICAA and ACS websites means in practice across participants, as even though they were given access to the resources, actual usage was not tracked. Was there any assessment if the ACS website was part of the usual/existing resources for participants? Were biweekly reminders and thank you emails sent to both groups or just the intervention arm? A lack of details makes it difficult to assess these potentially important components for interpretation of findings.	The use of TICAA and ACR websites has been tracked through the project website, but we did not include the data in this analysis. Both groups’ usages of ACR website were not significantly different. The use of TICAA was weekly checked with all the participants because the interventionists had weekly coaching/support sessions through the project website. Both groups have been sent weekly reminders to use TICAA and ACR website as well. We have added more details in the sections on settings and samples and enrollment and randomization.
Details of the recruitment settings are lacking, making assessment of potential selection biases (e.g. self-selection) and external validity difficult. The supplement outlines the original plan/protocol but does not detail the various support groups recruited from in the end, any deviations from the original plan, etc.	More details on the recruitment settings have been added. We followed the protocol. A total of 1,313 cancer support groups and communities/groups for Asian Americans were contacted, and 314 among them actually posted the study announcements. We did not track on how many were recruited through individual settings though.
The relative timing of where participants were in their treatment/survivorship journey is a critical piece of data compared across groups. Because most survivors were on average 3.5+ years out, the authors should comment on generalizability to women more recently diagnosed.	We did not collect the information on the participants’ treatment trajectories, but included the type and stage of breast cancer, time duration after the diagnosis of breast cancer, and treatment modalities that they went through. This point has been included in the limitation section.
Further elaboration on the cultural tailoring methodology would be helpful (e.g. were these informed only by the literature as stated or also through focus groups, community advisory panels, etc.).	We have conducted pilot studies including qualitative studies, which were the basis for cultural tailoring (e.g., cultural examples). This has been clarified on p. 9.
- A vague comment is made about adjusting for “covariates that demonstrated differences.” More detail about covariate selection and analytic adjustment is needed.	The details that were provided in the section on participants’ characteristics in the original submission have been added to the data analysis section as well. There existed no

	significant differences between the two groups except access to health care ($X^2=5.17$, $p<.02$) and age at immigration ($t=-2.34$, $p<.02$); these two variables were controlled in the subsequent data analysis process. Therefore, GEE was applied to confirm the differences in the changes of major variables between the two groups over time. During the process, two variables (access to healthcare and age at immigration) that showed differences in the homogeneity tests of the two groups were entered as covariates and adjusted for the subsequent analyses.
- There is a missed opportunity to statistically compare the full T0-T1-T2 trajectories for the various outcomes across intervention and control groups.	We totally agreed on the missed opportunity. Because of the page limitation, we intentionally exclude some sub-scale scores of individual primary outcome variables. We plan to submit separate papers with the detailed subscale scores under each primary outcome due to the page limit.
- The rationale for selecting an alpha of 0.10 is weak.	The sentence on selecting an alpha of 0.10 has been removed and the text has been revised accordingly.
Minor comments: - It is unclear from the “settings and samples” section if randomization was conducted within sub-ethnic groups (i.e. post-stratification) to achieve similar subethnic group sizes across arms. Also, it is unclear in the main text how participants were directed to the project website. - The data availability and sharing plan is informal with unclear criteria for approval of requests. - First sentence of the abstract is not a complete sentence. - Typo line 102: should read “The COVID-19 pandemic” or simply “COVID-19” - Reference needed for the second sentence of “limitations” in 305-308.	Yes, the randomization was done for each sub-ethnic group. This has been clarified in the section. More details on how the participants were directed to the project website have been added. Potential participants visited the project website after seeing the study announcement through online and offline cancer support groups and communities/groups for Asian Americans (the study announcement included the website address). More details on the data availability and sharing plan have been added. The first sentence has been modified. The spelling has been corrected. A reference has been added.
Reviewer #2	
The Authors reported the results of a randomized study supporting the use of a technology-based (virtual) information and coaching program in improving breast cancer survivors’ quality of life. The	Thanks.

study population was composed of Asian American women. Authors acknowledged strengths and limitations of the study but, overall, the discussion is well balanced.	
Reviewer #3	
I find it an important topic to study because we live in a diverse society and our information is not enough tailored to patients' needs.	Thanks.
The efficacy of culturally tailored programs has been studied before (e.g. recently doi: 10.1007/s13187-021-01970-y & 10.1177/1054773818755836). I miss here the specific reason why this needed to be studied, why this study is novel among all other studies that evaluated culturally tailored programs.	The program tested in doi: 10.1007/s13187-021-01970-y was for African American and Appalachian men. The program tested in doi: 10.1177/1054773818755836 was for 19 cancer survivors in South Korea. The population is totally different and the sample size for the second study (a similar population) was small (just 19). Both of them were not technology-based interventions (through computers or mobile devices). We have further clarified this in the introduction.
- Line 95: references 2-4: I would highly suggest to use more recent numbers (see doi: 10.3322/caac.21754)	The references have been updated again.
Line 101: reference are not peer reviewed articles and the second webpage could not be found. I do believe, however, that COVID has increased racism towards the Asian population, but I do not think this validates the need for this study, which was mostly done BEFORE the pandemic. The authors also mention COVID in their discussion and limitation, but the impact of COVID cannot be enourmous as the trial was finished in June 2020. And maybe the emphasis (during March and June 2020) would lie more on the anxiety of the possible impact of a COVID infection during cancer (treatment). It's therefore maybe better not to include data that was retrieved during the pandemic.	The final 6 months of the data collection was done during the COVID19 pandemic, but we included those recruited during the pandemic to have an adequate sample size. This point has been included as a limitation. The significance of the study does not depend on the COVID19 pandemic. Rather, the significance is on the novelty of a technology-based virtual intervention that is tailored for this specific population in need. We have revised the introduction to clarify the significance of the study.
- Lines 113-114: "reducing the unmet needs..." Is this the case, are there unmed needs, symptom distress and a lower quality of care; and if so, it is higher in comparison to other breast cancer patients? It is not because they seek less care, or report less symptoms, they are experiencing unmet needs. The authors need to elaborate on this matter a bit more to my opinion.	The clarification has been made in the introduction. In short, because of their cultural attitudes and values (e.g., stigma, stoicism), this specific population does not seek for adequate care or information, which makes them suffer from symptoms and pain that could be easily managed by using existing treatment/management modalities.
- Lines 132-141: repetition of the same information. The also do this between lines 281-291.	Any redundancies have been eliminated.

- Line 146: why did they send reminders. This probably also has influenced the behavior of participants.	Weekly reminders were to promote the participants' actual usages of the program (adherence), which is an elementary approach to boost adherence that has been frequently used in behavioral intervention studies. This has been clarified on p. 8. Reference: Horsch, Corine, Sandor Spruit, Jaap Lancee, Rogier van Eijk, Robbert Jan Beun, Mark Neerinx, and Willem-Paul Brinkman. "Reminders Make People Adhere Better to a Self-Help Sleep Intervention." Health and Technology 7, no. 2 (2017): 173–88. https://doi.org/10.1007/s12553-016-0167-x.
- Sociodemographics: unequal group sizes and the no differences were made in the group > high school education. I would expect to - especially for a study that looks at unmet needs etc, that they would include more groups for education: no education and primary school. Further, they did not ask how and where participants looked for information before the study.	This study is a technology-based intervention study using computers and mobile devices. We agree that our population tends to be highly educated because of the requirement for technology literacy. This is included as a limitation.
- I would suggest to ask for the calculation of their sample size! I have some doubts about this.	Again, we have re-calculated the sample size and added the following to the data analysis section: "In the sample size calculation, a sample size of 99 participants in each group (total N=198), with 3 repeated measurements obtained from each participant, would be adequately powered (80%) to detect a slope difference of 0.27, based on a two-group two-level hierarchical design. The calculation assumed a standard deviation of 1, a correlation of 0.1 between observations on the same subject, and an alpha level of 5%. The effect size was conventionally determined based on pilot studies."
- Lines 237-238: there were some participants lost in follow up, see fig 1 but this wasn't mentioned in the paper. So the total sample size was even lower.	This has been mentioned on p. 13. The data of the lost participants were also included in the data analysis (due to the intent-to-treat analysis), which is the reason that the total sample size is the same.
- Line 262: what is FACT-G?	FACT-G has been replaced with FACT-B (an error).
- Line 268: references are outdated.	We have tried to update the references one more time throughout the paper, but please note that some articles could not be replaced because there were no recent articles published on the topic/population after the last one.

- Discussion: cancer is a long term process with often many bad news conversations (diagnosis, treatment that does not work, ...) and different treatment approaches. So what does an intervention of 12 weeks mean? What is the added value of such a program compared to live interventions?	The discussion section has been extended with the two points that the reviewer indicated (the meaning of the 12 week intervention and the value of a technology-based intervention for this specific population)
---	---

REVIEWERS' COMMENTS

Reviewer #1 (Remarks to the Author):

The authors have adequately addressed my comments from the initial review.

Reviewer #3 (Remarks to the Author):

Dear authors,

I would like to express my gratitude for taking into account the comments that were made. I believe that the additions you made have significantly enhanced the clarity and significance of the paper/study.

I would like to give one minor suggestion: It would suggest to start the discussion section by providing a summary of the results, followed by a comparison with the findings from the existing literature.

Wishing you all the best!

Reviewer 1	
The authors have adequately addressed my comments from the initial review.	Thank you.
Reviewer 3	
I would like to express my gratitude for taking into account the comments that were made. I believe that the additions you made have significantly enhanced the clarity and significance of the paper/study. I would like to give one minor suggestion: It would suggest to start the discussion section by providing a summary of the results, followed by a comparison with the findings from the existing literature.	Thank you. A summary of the results has been added to the beginning of the discussion section and the findings are compared with those from the existing literature.